# Elemental Composition and Some Nutritional Parameters of Sweet Pepper from Organic and Conventional Agriculture

**DOI:** 10.3390/plants9070863

**Published:** 2020-07-08

**Authors:** Rosa Guilherme, Fernando Reboredo, Mauro Guerra, Sandrine Ressurreição, Nuno Alvarenga

**Affiliations:** 1CERNAS—Centro de Estudos de Recursos Naturais, Ambiente e Sociedade, Escola Superior Agrária de Coimbra, Instituto Politécnico de Coimbra, Bencanta, 3045-601 Coimbra, Portugal; rguilherme@esac.pt (R.G.); sandrine@esac.pt (S.R.); 2GeoBioTec, Departamento de Ciências da Terra, Faculdade de Ciências e Tecnologia, Universidade NOVA de Lisboa, Campus da Caparica, 2829-516 Caparica, Portugal; nuno.alavarenga@iniav.pt; 3LIBPHYS, Departamento de Física, Faculdade de Ciências e Tecnologia, Universidade NOVA de Lisboa, Campus da Caparica, 2829-516 Caparica, Portugal; mguerra@fct.unl.pt; 4Instituto Nacional de Investigação Agrária e Veterinária, I.P., UTI—Unidade de Tecnologia e Inovação. Avenida da República, Quinta do Marquês, 2780-157 Oeiras, Portugal

**Keywords:** nutritional characteristics, production mode, ripening stages, sweet peppers

## Abstract

The increasing demand of organic agriculture (OA) is based on the consumer’s belief that organic agricultural products are healthier, tastier and more nutritious. The effect of OA and conventional agriculture (CA) methods on the elemental compositions of green and red sweet peppers were studied. The highest concentrations of Ca, Cu, K and P occur in peppers from OA in both states of ripeness, with emphasis on Ca and K contents. Furthermore, the principal component analysis (PCA), points out to a clear separation, regarding concentrations, between peppers from OA and CA. The average fruit weight is higher in OA, 141 g *versus* 112 g in CA. Regarding productivity, CA reaches a value of 30.1 t/ha, 7% higher than the value observed for OA, i.e., 28 t/ha. Peppers from CA, exhibited greater protein content than those which originated from OA, regardless of the ripening stage, but not more ashes. Regarding nutritional ratios, the ripening stage and the production mode, can be important for an adequate choice regarding a more balanced Ca/P ratio, and the studied variety contained high Ca values ranging between 1009 and 1930 mg.kg^−1^. The PCA analysis also revealed that Mn and Fe are inversely correlated, confirming the importance of the Mn/Fe ratio evaluation in nutritional studies.

## 1. Introduction

Current agricultural systems have been mainly focused on yield rather than on balanced energy input/output ratios or on the sustainability of ecosystems and food production. It is well known that the intensive use of fertilizers, machinery and the agricultural practices as a whole, contribute to large greenhouse gas (GHGs) emissions [1]. Thus, farmers are facing a dilemma between pursuing their activities and maintaining or increasing their income, or adapting to climate change (CC) scenarios and reducing GHGs emissions.

Whatever the strategies for feeding the world in a more sustainable way, we currently assist a huge expansion of organic agriculture (OA) mainly based on an environmental awareness of the consumers and concerns with their food safety despite the criticism of some authors. According to the latest survey on OA, the organic farmland increased substantially, as well as the number of organic producers and organic retail sales [2]. In fact, although organic production currently accounts for only 1.4% of global agricultural land with approximately 70 million ha in 2017, it is responsible for a global organic market that has reached 97 billion US dollars in the same year (approx. 90 billion euros), involving almost three million producers worldwide.

The critics have long argued that organic agriculture is inefficient, requiring more land to yield the same amount of food [1]. However, when organic and conventional yields were compared using a new meta-dataset three times larger than previously used (115 studies containing 1071 observations, involving 38 countries and 52 crop species over a span of 35 years) it was observed that organic yields are only 19.2% lower than conventional yields, a smaller yield gap than previous estimates [3]. The same authors did not find significant differences in yields for leguminous and non-leguminous crops, nor for perennials and annuals, nor between the yield gaps for studies conducted in developed *versus* developing countries [3].

One of the most important functions of different agricultural production systems is to provide almost all essential mineral and organic nutrients to humans [4]. Additionally, there seems to be a widespread perception among consumers that organic farming results in products of higher nutritional quality [5]. It was concluded that organic products contained significantly more vitamin C, Fe, Mg and P, and significantly less nitrates and heavy metals when present, than products from conventional agriculture [6]. Most well-designed studies comparing nutrient density (milligrams of a given nutrient per kilogram of food) in organically and conventionally produced fruits and vegetables, show simple to moderately higher concentrations of nutrients in organic products [7].

Furthermore the evaluation of the mineral content of several edible products or even food supplements is paramount, since it allow us to monitor the enrichment or the poorness of essential elements to human nutrition and metabolism, or even detect possible contamination by heavy metals such as Cd, Pb, and Hg, among others [8,9,10]. In this context it is also useful to perform an acute characterization of the soil since potential redox, cation exchange capacity and particularly pH influences decisively affect the solubility and/or availability of the nutrients and the uptake by plants [11,12,13], beyond intrinsic aspects of the plants—that is, mainly related to species, varieties, development status and metabolic requirements, as well as to possible abiotic stresses that may occur [14,15].

Organic farming in Portugal started flourishing in the nineties, as the land and number of organic farmers had average annual growth rates above 20% and 40%, respectively [16]. In 2017, the Portuguese mainland area of organic production occupied 252,812 ha and was mainly dedicated to pastures and cattle fodder to feed livestock (72%), while approximately 26% of the area was dedicated to the production of foodstuff for direct consumption or for processing [17], and horticulture was included with only 1.2% of the area. Thus, the reduced offering of several highly appreciated products such as pepper, courgette, beans and lettuce for example, limits its sales growth in natural and gourmet food stores.

Pepper (*Capsicum annuum* L.) is one of the most important vegetables being produced and consumed in several countries worldwide. Pepper is cultivated in Portugal as a spring-summer crop in the open air. The aim of the present study is to evaluate the differences, regarding the elemental composition, and other nutritional characteristics, of two different cultures of sweet pepper (*Capsicum annuum* L.) using organic and conventional agricultural approaches and taking into account two different ripening stages. This information will contribute to increase the knowledge of the effect of the organic production method on the pepper´s nutritional value.

## 2. Results

### 2.1. Nutritional Characterization of Sweet Peppers

The elemental concentrations of sweet peppers, produced in organic agriculture (OA) and conventional agriculture (CA), and harvested in two gradual stages of ripeness (green and red) are presented in Table 1. The concentrations of macronutrients in sweet peppers in both modes of production and states of ripeness exhibits the following descending order: potassium (K) > phosphorus (P) > calcium (Ca) > and sulphur (S). Regarding the micronutrients, the descending order is: chlorine (Cl) > iron (Fe) > zinc (Zn) > manganese (Mn) > copper (Cu)—Table 1.

The K levels range between 3.27% and 4.02%, P between 1528 mg kg^−1^ and 2044 mg kg^−1^, Ca between 1009 mg kg^−1^ and 1930 mg kg^−1^, and finally S between 1232 mg kg^−1^ and 1644 mg kg^−1^. In what regard to micronutrients, Cl had the highest concentrations ranging from 2862 mg.kg^−1^ to 3892 mg kg^−1^, Fe from 64.7 mg kg^−1^ to 84.1 mg kg^−1^, Cu from 7.94 mg kg^−1^ to 11.8 mg kg^−1^, Zn from 15.7 to 25.0 mg kg^−1^, and finally Mn from 6.49 mg kg^−1^ to 22.5 mg kg^−1^ (Table 1).

In the case of K, P, Ca and Cu, the concentrations in peppers from organic agriculture (OA) were higher than those observed in peppers from conventional agriculture (CA), regardless of whether they were red or green. In fact, in the case of green peppers (OA) the concentrations of K, P, Ca and Cu were 1.29, 1.13, 1.02 and 1.91 times higher, respectively, than the equivalents from CA (Table 1). However, it should be noted that with the exception of Ca, the mean values were not significantly different (P ≤ 0.05), so the differences observed are not relevant, fitting the variability of a given population.

In the case of red peppers (OA), the concentrations of K, P, Ca and Cu were 1.12, 1.20, 1.17 and 1.26, times higher, respectively, than the equivalents observed for CA. Also, in this case, the mean values were not significantly different (P ≤ 0.05).

In relation to Mn and Zn, the highest concentrations were observed in peppers from CA, both green and red, compared to those derived from OA. In the case of green peppers (GP), there are significant differences between the averages observed in both modes of production and for both elements, whereas for red peppers (RP) there are also significant differences between the modes of production, but only for Mn.

Furthermore, Mn was the element where the greatest differences between the different cultivation modes were noted—for example, in green peppers from CA, the average concentration was approximately 3.0 times higher than that observed in peppers from OA. A similar situation occurred for red peppers (CA) that presented concentrations about 3.5 times higher than those of (OA).

In relation to Cl, Fe and S, there are differences that vary according to the ripening stage and the type of production. The concentrations of Cl, Fe and S in GP are higher in OA, the opposite is true for RP derived from CA. In a broader analysis, regardless of the type of ripening stage and the type of culture, there are no significant differences (P ≤ 0.05) with respect to the observed concentrations of Cl, Cu, K, P and S. Only in the case of Ca in organic GP (1930 mg kg^−1^) was a statistically significant difference was observed. A similar situation occurs for Zn, as well as for GP but from CA (Table 1).

Regarding the Ca/P ratio, it was observed that for organic GP this value was 0.98, while for conventional GP, this value was 0.66. In the case of red peppers, the Ca/P ratio was very close—0.64 vs. 0.61 for OA and CA, respectively. The Mn/Fe and Zn/Cu ratios are higher in CA, while the Fe/Cu ratio is similar in OA regardless of the ripening state, whereas in CA this ratio increases from 5.6 (GP) to 9.3 (RP) (Table 1).

Peppers from CA, exhibited more protein that those originated from OA, regardless of the ripening stage. Conversely, the ash content of RP and GP from OA is close, ranging from 5.87% to 6.38%, respectively, but higher than the values observed in CA, which range from 4.98% to 5.03%, for green and red, respectively. No significant differences were observed in the fiber content of GP from OA and CA (P ≤ 0.05); the same occurred with RP (Table 2).

The average fruit weight is higher in OA, with a value of 141 g per unit *versus* 112 g in CA. However, in terms of productivity, CA products a value of 30.1 t/ha slightly higher than the value observed for OA, i.e., 28 t/ha. In both methods (OA and CA), during the ripening process fruit weight decreases by approximately 23%.

### 2.2. Principal Component Analysis (PCA)

In order to jointly evaluate the influence of the production method and the state of ripeness, an analysis in main components (PCA) was carried out, using the data of OA and CA in two different ripening stages. Eight attributes (elements) were used, namely: P, Ca, Cu, Fe, Mn, K, Zn and S.

The first two main components explained the cumulative percentage of 65.25%:39.19% for the first component and 26.06% for the second. Only these components were significant, since they are those with an own value > 1. Thus, as they have an own value > 1, the first two components were defined as main components: the first had an own value of 3.1 and the second an own value of 2.1 (Table 3).

In order to understand the relative importance of each attribute in relation to each of the first two main components, the correlation coefficients between the attributes (original parameters) and the main components were determined (Table 4). The results of the 1st principal component (PC1), are explained, by P, Ca, Fe and K (with negative correlation values) and Mn (with positive correlation values), whereas the results of the 2nd principal component (PC2) are explained by Zn and S (with positive correlation values).

In Figure 1 it can be observed that the projection of the samples in the main plane, constituted by the first two components, is associated with the approximate projection of the attributes in the main plane.

The implementation of PCA, allow us to take the following results:1—The clear separation, with regard to concentrations, between peppers from OA and CA.2—Organic peppers tend to have higher concentrations of Ca, Fe, K and P and less Mn.3—In general, GP from both production methods have higher concentrations of S and Zn.4—In general terms, conventionally grown GP have a higher concentration of Mn and Zn than all the others.5—The Cu shows no significant correlation with any of the axes.

## 3. Discussion

Potassium is the most abundant element followed by P and Ca. This distribution is similar to that observed in two Spanish varieties of *C. annuum* [21]. Regarding micronutrient concentrations [21], the following order is observed: Fe > Mn > Zn > Cu. In the current study Zn ranks second while Mn is in the third position. The average concentrations of K detected by us are clearly higher than those found in peppers collected in areas as distinct as the Canary Islands(Spain) [22], Valencia(Spain) [23], Ethiopia [24], or South Korea [25].

The elementary composition of GP and RP from the Canary Islands [22], revealed a maximum K concentration of 250 mg 100^−1^ for RP and 199 mg 100^−1^ for GP. In the case of Ca the maximum value for RP was 33.2 mg 100^−1^ and 23.4 mg 100^−1^ for GP, while in the case of P, it was 40.0 mg 100^−1^ (RP) and 50.3 mg 100^−1^ (GP). The above-mentioned concentrations were clearly lower than those observed in the present study.

Other studies, however, report concentrations more closer to with those detected in the current work. When studying 17 varieties of peppers [23], it was observed that the average K concentration in GP was 2898 mg 100^−1^
*versus* 2155 mg 100^−1^ for RP. The same authors point out the average concentrations of P of 283 mg 100^−1^ for GP and 225 mg 100^−1^ for RP [23].

It was found that K concentrations are higher in GP, regardless of agricultural practices, while P concentrations are higher in RP, which partially agrees with the data [23]. The analysis of 12 varieties of RP grown in two different regions of South Korea [25], showed that the average levels of Fe in the Imsil (IS) and Youngyang (YY) regions were 6.1 mg 100^−1^ and 6.6 mg 100^−1^ respectively. Regarding Ca, the same authors point out the mean values of 114 mg 100^−1^ in IS and 104 mg 100^−1^ in YY, while for K the mean concentrations were 2821 mg 100^−1^ (IS) vs. 4125 mg 100^−1^ (YY), levels quite concordant with those detected in the current work.

The variability of the data is also documented in an extensive study, regarding vitamins A and C, folate, and capsaicin concentrations of different pepper types, emphasizing the huge variations within species, varieties, color, and geographic locations [26]. Other details can also explain the variability. While in some works [21,25] the analysis is carried out with dehydrated samples (in our case reduced to powder) in others [22,23], it is not clear whether the samples for analysis are weighed without dehydration, which occurs *a posteriori* when determining the elemental composition.

Additionally, the effect of fruit maturation on quantitative changes in elemental composition and organic compounds must also be taken into account. For example, in *Capsicum annuum, Capsicum frutescens,* and *Capsicum chinese*, the concentration of carotenoids, flavonoids, total soluble reducing equivalents, phenolic acids, ascorbic acid, and antioxidant activity generally increased with maturity [27], although in two common Spanish *Capsicum* varieties, none influence on elemental concentration was observed as a function of the ripening stage [21].

The protein content of two Spanish *Capsicum* varieties ranged between 11.37% and 12.02% regardless of the ripening stage [21], while the variation within three *Capsicum* varieties from Ethiopia was between 8.7% and 11.8% [24]. The same authors observed that the ash content varied between 4.27% and 4.68% [21] and between 5.3% and 7.3% [24]. The mentioned values are in good agreement with our values, although other studies reported much higher values of both protein and ashes [25].

The application of N, P, K and S fertilizers increased crop yield and protein concentration in cereals and pulses, and concentration of essential amino acids and vitamins in vegetables. However, excessive fertilizer use, especially N fertilizer, can result in undesirable changes such as increases in nitrate, titratable acidity and acid to sugar ratio, while decreasing the concentration of vitamin C, soluble sugar, soluble solids, and Mg and Ca in some crops [4], which agrees with our results since fiber and protein contents are higher in CA peppers.

Also, titratable acidity (TA) is similar in RP, regardless of the type of production, thus indicating that TA increases during the ripening process in parallel with a decrease of pH. A similar situation occurs for total soluble solids (TSS). In this context, it does not seems to us that an excessive fertilizer use had occurred [4], and the differences observed are mainly related to the variety itself since the agro-environmental factors are similar, except the fertilization process, although other authors point out to an inverse relationship between TA and pH—during the ripening of cactus *Cereus peruvianus* fruit, the TA decreased and the pH increased [28], which also occurred during the ripening of blackberry fruit [29], they make the assumption more cristal-clear.

Organic foods (vegetables, legumes and fruit) were found to have a 5.7% higher content of vitamins and minerals than their conventionally grown counterparts with emphasis on P levels [30]. These results were explained by the hypothesis of accelerated growth, as a result of conventional agricultural methods, that down-regulates the synthesis of carbon-containing metabolites, such as ascorbic acid [30]. In our case both P and ash levels are higher in OA peppers.

The average weight of each fruit, regardless of the ripening stage and the production mode, varied between approximately 100 g and 150 g, a very low weight when compared with the range of 350 g and 500 g, observed in two different varieties [21], emphasizing the great variability among varieties, beyond the production mode and substrata characteristics. Furthermore, our production around 30 t/ha is greater than the world average production of chilies and peppers (*Capsicum* spp.) in 2018, which is 18.5 t/ha but below the average of EU28 in the same year, which is 41.2 t/ha. Despite China having an average of 23.6 t/ha, it is responsible for approximately half of the world production [31].

Regarding the Ca/P ratio, lower values (0.52 and 0.51) were observed for two varieties of *C. annuum* from Nigeria [32], whereas the Ca/P ratio values in a Spanish *C. annuum* variety decreases during the ripening process, from 1.29 (GP) to 0.61 (RP) [22]. *Capsicum annuum* is one of the plant species with the lowest levels of Ca (<8.7 mg/100 g), whereas P is present in vegetables in the range of 16.2–437 mg/100 g [33], thus a prevalence of P over Ca is common.

The Ca/P ratio is important for human nutrition Excessive intake of P *per se* can be harmful to bones through increased parathyroid secretion, but the adverse effects on bone mass increase, when dietary Ca intake is clearly low [34]. In many countries, P intake is abundant, while Ca intake does not meet the recommendations, so it is difficult to achieve an optimal proportion of Ca/P in the diet [34]. Thus, the ripening stage and the production mode of peppers, can be important for an adequate choice regarding a more balanced Ca/P ratio.

The negative interaction of metal ions is one of the main dietary factors that causes low bioavailability of these nutrients. These include Na-K, Ca-Mg, Mn-Fe, Fe-Cu, and Zn-Cu, the latter being the most significant in human nutrition due to the negative effect of excess Zn on Cu bioavailability, i.e., when the first metal of each pair is in excess and the other is at the lower limit of requirement [35]. In our case the Zn/Cu ratios range between 1.63 and 2.34 indicating that a good balance exists between these two elements without compromising Cu availability.

In regard to Mn-Fe interactions, it was observed that Mn inhibited iron absorption both in solutions and in a hamburger meal in human trials, most probably due to similar absorptive pathways [36]. In the same experiment with Zn, no inhibitory effect was observed, suggesting different pathways for the absorption of Zn and Fe. In our case the Mn/Fe ratios did not exceed 0.329, indicating that Mn is present in very low amounts in sweet peppers. Furthermore, in the principal component analysis, Mn and Fe are inversely correlated in component 1 (PC1)—one positively (Mn: 0.62) and the other negatively (Fe: −0.75).

Adequate Cu nutritional status is necessary for normal Fe metabolism and red blood cell formation, indicating an interconnection between Cu availability and Fe metabolism in humans [37,38]. For example, infants fed a high-iron formula absorbed less Cu than infants fed a low-iron formula, suggesting that high Fe intakes may interfere with Cu absorption [39]. The sweet pepper Fe/Cu ratios range between 5.58 and 9.27 indicating a prevalence of Fe over Cu.

One may argue that the search of ideal elemental ratios is not worthwhile because there is a difference between the dietary ratio and the blood (or serum) ratio, the latter more relevant in health and disease [40]. However, the load of a particular element in a certain foodstuff may well give us an indication of balance/imbalances, without losing the focus on the interactions and bioavailability and as recently referred, the utilization of dietary micronutrient ratios in nutritional studies might well be more informative than focusing on a single nutrient [41,42].

For example, a high intake of Fe, especially in combination with high Mn intake, may be related to risk for Parkinson´s disease [43], thus, consumers must avoid foodstuffs particularly rich in both elements. The awareness of these interactions, combined with a balanced evaluation of the dietary intake, could lead to more effective strategies to improve micronutrient status [44], although currently, the biofortification of staple foods [45,46,47] and the consumption of food supplements [10] stay ahead, despite the negative effects of Fe supplementation on indices of Zn and Cu status and of Zn supplementation on Fe and Cu status that have been reported [44].

The production of *Capsicum* species is cheap and easy, therefore, integrating a pepper-rich diet in our daily meals can be helpful in alleviating micronutrient deficiency, especially in poor households in developing countries [48] thus preventing the appearance of chronic diseases.

## 4. Materials and Methods

### 4.1. Plant Cultivation

Two producers of sweet peppers (*Capsicum annuum* L.), located in the central region of Portugal (Coimbra) were selected with the following coordinates: Organic agriculture—40°13′4.71″ N (latitude); 8°26′58.28″ W (longitude). Conventional agriculture—40°13′14.00″ N (latitude); 8°28′29.22″ W (longitude). The organic producer, followed the European Commission Guidelines (Council Regulation (EC) nº 834/2007 of 28 June 2007) whereas the other producer, had no limitations on the use of fertilizers and pesticides.

Sweet pepper seedlings, from the Entinas variety, were put in the soil in May 2018 (Table 5), and grew under open field conditions until September, i.e., it took approximately four months between planting and harvesting sweet peppers. Each field had 800 m^2^ and the density of plantation was 22,222 plants/ha corresponding to a distance between the lines of 0.75 cm and between plants of 0.60 cm.

The distance between experimental fields was approximately 950 m and the meteorological conditions registered in the area by the Agrarian School of the Polytechnic Institute of Coimbra (where the fields are located) are described as follows: temperature (°C)—maximum (22.3; 24.2; 26.1; 32.0 and 31.5), average (16.4; 19.0; 20.9; 22.7 and 21.4) and minimum (11.0; 15.1; 16.8; 15.7 and 15.0); evapotranspiration (mm)—(98.7; 92.5; 109.6; 123.6 and 95.0); relative air humidity (%)—(78.8; 84.6; 82.2; 74.4 and 77.4); rainfall (mm)—(37.6; 98.8; 7.8; 0.0 and 1.2). The values are presented in a sequential manner, i.e., the first one is from May, the second from June, the third from July, the fourth from August and finally, the fifth from September.

Soil characterization can be indicated as follows: OA (pH 6.4; organic matter 1.5%; extractable P—mg P_2_O_5_ kg^−1^, 105; extractable K—mg K_2_O kg^−1^, 156); CA (pH 6.0; organic matter 1.8%; extractable P—mg P_2_O_5_ kg^−1^, 181; extractable K—mg K_2_O kg^−1^, 134).

In both fields, a drip irrigation system was installed and the nutritional requirements were supplied by horse manure, in OA, and by chemical fertilizers, in CA. The water for irrigation came from the Mondego River (pH 7.34; EC 0.10 mS/cm^−1^; SAR 0.8 meq L^−1^). The drip irrigation system had a flow rate of 2.66 L/s per emitter for approximately 20 min with irrigation intervals between 2 or 3 days, depending on evapotranspiration conditions and precipitation occurrence.

Regarding fertilization of CA, it was made with chemical fertilizers: before planting (700 kg/ha) with: 7% N, 14% P_2_O_5_, 14% K_2_O, 3% CaO, 2% MgO, 9% SO3 and 0.02% B; and at flowering, late July, (300 kg/ha) with 27% N and 4% CaO. In OA, horse manure was incorporated into the soil before planting (29.6 t/ha) with 5.8 g of N/kg^−1^, 2.8 g of P_2_O_5_/kg^−1^ and 5.3 g of K_2_O/kg^−1^. The weeds control was performed with herbicide (CA) and with mechanical and manual methods at OA. In CA, two weeks before crop planting a selective weed herbicide was applied (glyphosate, 4–5 L/ha).

The phenological development stages of sweet pepper from OA and CA during 2018, can be observed in Table 5. It is clear that peppers from CA initiate in advance the leaf development, the flowering and the development of the fruit, expressed as percentage of plants where these stages were observed, which is probably related to the use of fertilizers and particular characteristics of the soils, despite being transplanted into the soil five days after organic plants.

### 4.2. Sample Collection

In the middle of September 2018, from each producer (*organic* and *conventional*) sweet peppers from each producer were harvested on the same day at two different maturation stages, corresponding to green and red colours. Four independent batches with approximately 2 kg each (two for each production mode), were collected. Each batch containing 15 sweet peppers was then divided into five groups of three units. From each group one pepper was removed to perform moisture and elemental determination.

A total of 60 peppers were used for analysis. Forty of them were washed, cleaned, air-dried, and further stored under refrigeration (~4 °C), until analysis (protein, TA, TSS-°Brix, fiber, ash). The determination of firmness, was made prior to storing samples in the refrigerator. The remaining 20, previously washed and cleaned, were cut in slices and the edible part was separated from non-edible (seeds). Samples were weighed (fresh weight) and dried at 65 °C until constant weight. After the determination of dry matter and moisture, powder reduced samples in an agate mortar were used for XRF analysis.

Productivity (t/ha) was derived from 30 selected plants along the harvesting phase. Approximately 260 peppers from OA and 370 from CA were collected and weighed after careful rinsing with distilled water.

### 4.3. Nutritional Composition

The methodologies of the Association of Official Analytical Chemicals [49] were used to determine chemical properties of the dried pepper samples, namely, crude fiber (method 930.10), ash (method 930.05) and crude protein (method 978.04). Additional detailed information is found in [50].

Regarding the firmness the maximum penetration force (N) was evaluated with a HD plus texture analyzer (Stable Microsystems, Godalming, UK). The evaluation was made by penetration with a 2 mm cylinder probe, with a 5.0 kg (50 N) charge cell, and with a test speed of 1.0 mm/s and 10 mm length [51].

The titratable acidity (TA) was determined by titrimetric analysis, with a NaOH solution (0.10 mol/L). Approximately, 10 g of each sample (previously ground) were mixed with 50 mL of water and put on heating under reflux for 30 min. Then, the resultant solution was transferred to a glass balloon of 100 mL and, after filtration, a precise volume (20 mL) was transferred to a beaker with a stirrer. Then, the pH of the solution was monitored continuously in order to obtain the titration curve. The values were expressed on mg citric acid/100 g fresh weight [52].

The pH values were evaluated using a Crison-Micro pH 2002 (Crison, Barcelona, Spain) potentiometer. The solution obtained for the acidity determination (after filtration) was also used to measure total soluble solids (TSS) contents (°Brix), at 20 °C, in an ATAGO refractometer (Saitama, Japan) [52].

### 4.4. XRF Preparation and Analysis

The fine powder of dried pepper samples was pressed for 2 min under 10 tons in order to make a cylindrical pellet with a diameter of 20.0 ± 0.5 mm and a thickness of 1.0 ± 0.5 mm. This pellet was then glued onto a mylar sheet in a plastic frame and placed directly onto the X-ray beam for analysis.

All of the elemental quantifications were obtained using the micro-energy dispersive X-ray fluorescence (µ-EDXRF) system (M4 TornadoTM, Bruker, Germany). This commercial spectrometer consists of an air-cooled micro-focus side window Rh-anode X-ray tube, powered by a low-power HV generator. The system features a poly-capillary X-ray optic, which allows a beam spot size of around 22 µm for Rh Kα. In all of the measurements, the X-ray generator was operated at 50 kV and 100 µA without the use of filters, in order to enhance the ionization of low-Z elements without compromising the peak-to-background ratio for medium-Z elements such as Mn, Fe, Cu and Zn [53]. Detection of the fluorescence radiation is performed by an energy-dispersive silicon drift detector, XFlashTM, with a 30 mm^2^ sensitive area and an energy resolution of 142 eV for Mn Ka.

In order to obtain an average spectrum that is representative of the whole pellet, elemental maps were acquired with a pixel spacing of 15 µm and a measuring time of 6 ms per pixel. Quantification of the spectra of the obtained maps was performed with the fundamental parameters method of the built-in ESPRIT software and the recovery rate was checked against a set of standard reference materials—orchard leaves (NBS 1571), poplar leaves (GBW 07604) and tea leaves (GBW 07605).

The achieved detection limits with this setup can be seen in reference [54], and are around 5, 4, 55, 35, 4 and 15 µg/g, for S, Cl, K, Ca, Mn and Fe, respectively. For Cu and Zn, the detection limit is 2 µg/g. The Na and Mg levels, were not assessed, since elements with low atomic number (<13) were not quantified by µ-EDXRF, unless present in large concentrations. Thus, they are referred as below the detection limits.

### 4.5. Statistical Analysis and Control Assurance

Statistical analysis of the data was performed with the SPSS Statistics 18 program, through an analysis of variance (ANOVA) and the F-test. A value of P ≤ 0.05 was considered to be significant. All analyses were made in quintuplicate. Analytical accuracy was verified using replicate determinations and Standard Reference Materials, as referred above, with percentages of recovery ranging between 92% and 99%. Principal component analysis was applied to observe any possible clusters within analyzed sweet peppers from OA and CA in two different ripening stages (green and red) with the STATISTICA (data analysis software system), version 12.

## 5. Conclusions

The highest concentrations of Ca, Cu, K and P occur in peppers from OA regardless the state of ripeness. Furthermore, the PCA highlights out to a clear separation, regarding concentrations, between peppers from OA and CA.

In general terms, green peppers from both production methods have higher concentrations of S and Zn, but they are also a good source of Ca and K. In what regard to nutritional ratios it is important to emphasize that, the ripening stage and the production mode, can be important for an adequate choice regarding a more balanced Ca/P ratio. In the case the Mn/Fe, Mn is present in very low amounts and does not seem to influence the Fe absorption. The adequate selection of micronutrient-rich foods in the diet can alleviate micronutrient deficiency, thus preventing the appearance of chronic diseases.

## Figures and Tables

**Figure 1 plants-09-00863-f001:**
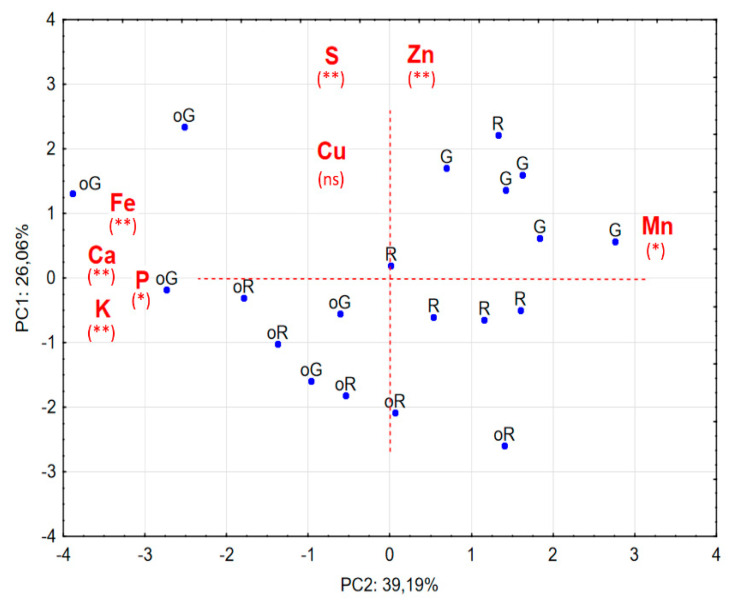
Analysis of main components: PC1 vs. PC2 projection of samples (*n* = 5). The most important variables for the definition of the two components are shown in each axis, indicating the direction in which each element grows. Five samples of each of the peppers, are presented in the projection: peppers obtained in conventional production mode, red (R) and green (G) and peppers obtained in organic production mode, oR and oG. * Moderately significant correlation values between the element and the PC; ** strongly significant correlation values between the element and the PC; ns = non-significant correlation between PCs.

**Table 1 plants-09-00863-t001:** Average concentrations of sweet peppers originated from organic and conventional agriculture.

Elements	Green Peppers		Red Peppers	
	Organic	Conventional	Organic	Conventional
K	4.02 ± 0.299 a	3.55 ± 0.290 ab	3.65 ± 0.276 ab	3.27 ± 0.345 b
Ca	1930 ± 144.8 a	1009 ± 97.10 b	1314 ± 365.8 b	1041 ± 203.3 b
P	1974 ± 394.7 a	1528 ± 208.6 a	2044 ± 281.38 a	1704 ± 313.2 a
Fe	84.1 ± 10.4 a	64.7 ± 5.4 b	68.0 ± 13.9 ab	73.6 ± 8.5 ab
Zn	19.2 ± 2.27 b	25.0 ± 2.43 a	15.7 ± 1.87 b	18.6 ± 1.82 b
Mn	7.37 ± 2.07 b	21.3 ± 4.51 a	6.49 ± 1.92 b	22.5 ± 11.5 a
S	1644 ± 329.9 a	1518 ± 216.3 a	1232 ± 162.2 a	1566 ± 204.5 a
Cu	11.8 ± 4.21 a	11.6 ± 2.21 a	9.32 ± 0.99 a	7.94 ± 4.33 a
Cl	3892 ± 630.9 a	3250 ± 578.5 a	2862 ± 959.2 a	2984 ± 437.1 a
Ca/P	0.98	0.66	0.64	0.61
Mn/Fe	0.087	0.329	0.095	0.306
Fe/Cu	7.13	5.58	7.30	9.27
Zn/Cu	1.63	2.16	1.68	2.34

The mean values are expressed in % (K) and mg kg^−1^, on a dry weight basis (remaining elements) ± standard deviation; *n* = 5; the mean values in the same row, followed by common letters, are not significantly different at 0.05 significance level.

**Table 2 plants-09-00863-t002:** Nutritional characterization of sweet peppers from organic and conventional agriculture.

	Green Peppers		Red Peppers	
	OA	CA	OA	CA
Fruit weight	159 ± 37.7 a	127 ± 16.1 ab	123 ± 13.8 ab	97.4 ± 18.9 b
Moisture *	93.9 ± 1.30 a	92.5 ± 0.60 a	91.8 ± 0.75 a	90.3 ± 0.43 b
Protein *	10.3 ± 0.62 b	11.7 ± 0.56 a	9.17 ± 0.49 b	11.9 ± 0.42 a
Ash *	6.38 ± 0.58 a	4.98 ± 0.32 b	5.87 ± 0.28 a	5.03 ± 0.27 b
Fiber *	11.4 ± 0.93 a	11.6 ± 0.74 a	9.23 ± 1.00 b	10.3 ± 1.06 ab
Firmness **	12.1 ± 2.3 a	7.9 ± 0.9 b	9.2 ± 1.4 ab	7.9 ± 1.2 b
pH **	6.2 ± 0.1 a	6.0 ± 0.5 a	5.0 ± 0.1 b	5.2 ± 0.1 b
TSS(°Brix) **	3.8 ± 0.3 c	4.4 ± 0.5 c	5.8 ± 0.7 b	7.6 ± 0.5 a
TA **	63 ± 4.0 b	67 ± 9.0 b	162 ± 16.0 a	171 ± 18.0 a
Organic Agriculture			Conventional Agriculture	
Productivity	28.0		30.1	

(*) The mean values are expressed in percentage (%) ± standard deviation; fruit average weight (g) ± standard deviation; *n* = 5; productivity (t/ha) was derived from 30 selected plants along the harvesting phase. (**) Firmness, pH, total soluble solids (TSS) and titratable acidity (TA) expressed as mg citric acid/100 g fresh weight, from ref. [18]. The mean values in the same row, followed by common letters, are not significantly different at 0.05 significance level.

**Table 3 plants-09-00863-t003:** Principal component analysis (PCA). Percentage of variance for each component (initial eigenvalues) and correlation coefficients (component matrix) of each variable with component 1 (PC1) and 2 (PC2).

Principal Component	Eigenvalue	Total Variance (%)	Cumulative Eigenvalue	Cumulative (%)
1	3.13	39.19	3.13	39.19
2	2.08	26.06	5.22	65.24
3	0.89	11.09	6.11	76.33
4	0.63	7.93	6.74	84.26
5	0.44	5.51	7.18	89.76
6	0.33	4.17	7.52	93.94
7	0.25	3.17	7.77	97.11
8	0.23	2.89	8.00	100.00

**Table 4 plants-09-00863-t004:** Correlation coefficients between attributes (initial variables) and PC1 and PC2.

	Components	
Attribute	PC1	PC2
P	−0.68 *	−0.02
Ca	−0.82 **	0.09
Cu	−0.37	0.55
Fe	−0.75 **	0.32
Mn	0.62 *	0.57
K	−0.86 **	−0.19
Zn	0.26	0.83 **
S	−0.29	0.79 **

* Values considered moderately correlated with PC (0.6 < r < 0.7); ** Values considered strongly correlated with PC (r > 0.7) following the classification used previously [19,20].

**Table 5 plants-09-00863-t005:** Phenological development stages of sweet pepper from organic (OA) and conventional (CA) agriculture (% of plants).

Stage Development	PM	16/May	21	28	04/Jun	11	18	25	02/Jul	09	16	23	30	06/Aug	13	20	27	03/Sept	10	17	24t
Transplanting	OA																				
	CA																				
Leaf Development	OA				**≤10%**	**10–25%**	**25–50%**	**50–75%**	**≥75%**	**≥75%**											
	CA				**≤10%**	**10–25%**	**50–75%**	**50–75%**	**≥75%**	**≥75%**											
Flowering	OA								**≤10%**	**10–25%**	**25–50%**	**50–75%**	**≥75%**	**25–50%**	**10–25%**						
	CA								**≤10%**	**25–50%**	**25–50%**	**50–75%**	**≥75%**	**25–50%**	**10–25%**						
Development of Fruit	OA										**≤10%**	**10–25%**	**25–50%**	**50–75%**	**≥75%**	**≥75%**	**50–75%**	**25–50%**			
	CA										**≤10%**	**25–50%**	**25–50%**	**50–75%**	**≥75%**	**≥75%**	**50–75%**	**25–50%**			
Ripening of Fruit	OA													**≤10%**	**10–25%**	**25–50%**	**50–75%**	**≥75%**	**≥75%**	**≥75%**	**≥75%**
	CA													**≤10%**	**25–50%**	**25–50%**	**50–75%**	**≥75%**	**≥75%**	**≥75%**	**≥75%**

PM = Production Mode.

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
