# Peer review of "Elemental Composition and Some Nutritional Parameters of Sweet Pepper from Organic and Conventional Agriculture"

_plants, 2020, doi:10.3390/plants9070863_

Round 1
Reviewer 1 Report
Please, correct the title into: '... organic and conventional agriculture' NOT 'agricultures'
It is not clear the message provided in the lines 57-61. Please, carefully check the references and transform this sentence, in order to be understandable by the readers.
The main issue is to compare conventional and organic production systems for pepper plants. The topic is clearly interesting and provides some novelty.
However, the paper is not very well-written (the discussion is descriptive and sometimes monotonous for the readers; this is why it does not provide high interest for the reader, although the topic is interesting).
Generally, the paper is technically correct (M&M), it provides some novelty, but it is not one of the most interesting manuscripts I have ever read-this is why I believe it will not be very appealing for the readers.
Author Response
COVER LETTER
Manuscript ID: plants-848192
REVIEWER 1
Remark 1
Please, correct the title into: '... organic and conventional agriculture' NOT 'agricultures'
Your remark was corrected
Remark 2
It is not clear the message provided in the lines 57-61. Please, carefully check the references and transform this sentence, in order to be understandable by the readers.
These lines were rewritten and we hope the message is now clear to the readers (lines 55-61).
Remark 3
The main issue is to compare conventional and organic production systems for pepper plants. The topic is clearly interesting and provides some novelty. However, the paper is not very well-written (the discussion is descriptive and sometimes monotonous for the readers; this is why it does not provide high interest for the reader, although the topic is interesting).
Your remark was carefully evaluated and in our opinion the discussion was less descriptive, despite new arguments were added. For example in the last draft the Discussion contained 127 lines and the current one, only 110 lines including 12 new lines (248-55; 264-67).
Author´s Concluding Remark
Some data related to productivity, fruit weight and other parameters was addressed in the Abstract, as requested by one of the reviewers (lines 27-30)
The phenological development stages of sweet pepper from OA and CA during 2018 was included, see lines 338-42 and Table 5 (lines 358-60).
Best Regards
Fernando H. Reboredo
Reviewer 2 Report
The authors investigated the differences between elemental and nutritional composition of sweet pepper (Capsicum annum L) cultivars, using organic and conventional agricultural approaches during two gradual stages of ripening. Having read the manuscript, I can recognize that the information included is sound, and the topic is within the scope of the journal, but I have found some important gaps that make this manuscript not suitable for publication, especially because it does not meet the requirements of a q1 journal as the case of Plants-MPDI. Some portions of the article are wordy and quite repetitive, potentially the authors could make the paper more concise in the future revision. Methodology used is appropriate and including PCA analysis (please describe this in M&M as other aspects that they will be omitted in the section). To my mind, the main drawback for being publishing is the few data presented. Indeed, the paper just describe dry weight/water content and nutritional elements obtained by XRF. Do not explain vegetative growth (which was alter by the agriculture approach used), production, fruit quality etc. Additional data is needed since the results do no support well the conclusions presented. Moreover, some sentences need to be rephrased in order to be understandable. Finally, the novelty of the effects, if any, should be point out throughout the manuscript.
Author Response
COVER LETTER
Manuscript ID: plants-848192
REVIEWER 2
Remark
Some portions of the article are wordy and quite repetitive, potentially the authors could make the paper more concise in the future revision. Methodology used is appropriate and including PCA analysis (please describe this in M&M as other aspects that they will be omitted in the section).
- Your remark was carefully evaluated and in our opinion the discussion was less descriptive and repetitive, despite new arguments were added. For example in the last draft the Discussion contained 127 lines and the current one, only 110 lines including 12 new lines (248-55; 264-67).
- In the MM section a brief but concise description about PCA analysis was introduced
Remark
To my mind, the main drawback for being publishing is the few data presented. Indeed, the paper just describe dry weight/water content and nutritional elements obtained by XRF. Do not explain vegetative growth (which was alter by the agriculture approach used), production, fruit quality etc. Additional data is needed since the results do not support well the conclusions presented. Moreover, some sentences need to be rephrased in order to be understandable. Finally, the novelty of the effects, if any, should be point out throughout the manuscript.
- Your remark about few data is not correct since we can compare this type of study and realize that the overwhelming majority of similar studies, does not evolve a PCA analysis, nor an approach related to nutritional ratios. Furthermore, our average values are based on five replicates when it is more common to have only, three replicates. It is easy to read some of the works quoted in our study and confirm this statement.
- The discussion about nutritional ratios is an up-to-date problem often ignored by persons like me – a plant biologist, despite being faced in the future as a tool to improve nutrient deficiencies. More than use Caco Cells experiments we must know in advance the elemental content of a certain foodstuff and the particular ratios. The information extracted from the ratios was fully justified in the PCA analysis.
- The phenological development stages of sweet pepper from Organic Agriculture and Conventional Agriculture during 2018 was included (see lines 338-42) as well as a new Table 5 (lines 358-60) and the discussion was enriched with new data, as referred above. It is always our goal to make the manuscript clear and comprehensible, despite the divergent opinions that might exist.
Best regards
Fernando H. Reboredo
Reviewer 3 Report
Major comments
- The title of the work ‘Elemental composition and nutritional importance of sweet pepper from organic and conventional agricultures’- includes words ‘nutritional importance”. But there are no data in the manuscript about the most valuable biochemical characteristics: vitamin C, phenolics and carotenoids content though the authors discuss the importance of these parameters (see lines 247-251 and 257-262). In these respect the title seems not to fully reflect the content of the work and some changes seem to be necessary. May be: ‘Elemental composition and several biochemical characteristics…’ Not sure, what decision will be the best. It should be also mentioned that Abstract also does not contain any information on ‘biochemical characteristics investigated and at the ent of introduction (line 90) ‘other nutritional characteristics’ are mentioned.
- Table 2 presents two close parameters: dry matter and moisture content, the latter being calculated as a difference between 100 % and Dry matter value. One should be deleted.
- While discussing the importance of ratios approach and indicating “titratable acidity/carbohydrates (Brix)’ ratio (line 273) –this ratio was not used in the discussion
- While reading the experimental part a question arises: are there any differences in the amount of seeds in sweet pepper fruit grown under organic and conventional practice technique?
- According to literature data protein content in sweet pepper is about 2%. The data of Table 2 indicate 9-12%. Necessary to be explained
Minor comments
- The results of elements content are expressed in mg.kg-1- it will be better to add everywhere ‘d.w.’ because the reader may indicate it fact only in experimental part
- Table 2 ‘productivity’ is given at the stage of red fruit- so better to put these data in the two right columns: ‘red pepper OA and red peppers CA
- Line 209 ‘In the projection are presented the 5 samples’ should be changed to : Five samples are presented in the projection’
- Line 297:’ Excessive intake of P per se can be harmful to bones through increased parathyroid secretion, but the adverse effects on bone mass increase, when dietary Ca intake is clearly low.’ – a reference is desirable
- Line 312 ‘ratios does not exceeded’ should be changed to “ did not exceed’
Author Response
COVER LETTER
Manuscript ID: plants-848192
REVIEWER 3
Remark
The title of the work ‘Elemental composition and nutritional importance of sweet pepper from organic and conventional agricultures’- includes words ‘nutritional importance”. But there are no data in the manuscript about the most valuable biochemical characteristics: vitamin C, phenolics and carotenoids content though the authors discuss the importance of these parameters (see lines 247-251 and 257-262). In these respect the title seems not to fully reflect the content of the work and some changes seem to be necessary. May be: ‘Elemental composition and several biochemical characteristics…’ Not sure, what decision will be the best.
- I fully agree with your comment since the most valuable nutritional characteristics are not addressed. In this context a small change in the title was performed.
-Regarding your comment about the discussion of vitamin C, carotenoids. But what is relevant in those paragraphs is the huge variability found within species, varieties, geographic location. We regret that the authors did not give us information about elemental data, but we might well assume that the variability will be huge, also.
Remark
It should be also mentioned that Abstract also does not contain any information on ‘biochemical characteristics investigated and at the end of introduction (line 90) ‘other nutritional characteristics’ are mentioned.
- We delete a few lines in the Abstract about macro and micronutrient data and as requested we added new lines about nutritional parameters and productivity (lines 27-30). Your comment is completely justified.
Remark
Table 2 presents two close parameters: dry matter and moisture content, the latter being calculated as a difference between 100 % and Dry matter value. One should be deleted.
- We delete the dry matter value
Remark
While discussing the importance of ratios approach and indicating “titratable acidity/carbohydrates (Brix)’ ratio (line 273) –this ratio was not used in the discussion.
- This issue was addressed in lines 248-255.
Remark
While reading the experimental part a question arises: are there any differences in the amount of seeds in sweet pepper fruit grown under organic and conventional practice technique?
- We had no collected data about that issue
Remark
According to literature data protein content in sweet pepper is about 2%. The data of Table 2 indicate 9-12%. Necessary to be explained
- As seen in lines 236-241, several data regarding the protein content was indicated “The characterization of both green and red peppers from OA and CA gave us percentage values of protein, ash and fiber which in general agree with the values referred by other authors. The protein content of two Spanish Capsicum varieties ranged between 11.37% and 12.02% regardless of the ripening stage [21] while the variation within three Capsicum varieties from Ethiopia was between 8.7% and 11.8% [24]. The same authors observed that the ash content varied between 4.27% and 4.68% [21] and between 5.3% and 7.3% [24]. The mentioned values are in good agreement with our values although other studies reported much higher values of both protein and ashes [25]”.
I have never seen such low protein content. It is interesting to see the link below which point out to percentage values close to ours.
https://fdc.nal.usda.gov/fdc-app.html#/food-details/787811/nutrients
They indicate a protein content of 0,86 g on a 100g fresh weight basis. If we take into account the moisture, always above 90%, we will reach a value higher than 10%.
Remark
The results of elements content are expressed in mg.kg-1- it will be better to add everywhere ‘d.w.’ because the reader may indicate it fact only in experimental part
- In the Table 1 we add on a dry weight basis, i.e., mg.kg-1, on a dry weight basis
Remark
Table 2 ‘productivity’ is given at the stage of red fruit- so better to put these data in the two right columns: ‘red pepper OA and red peppers CA
- We introduce in the Table a splitting area to distinguish the productivity data from previous data. The productivity as whole, does not distinguish green and red peppers, only organic and conventional production. The data was obtained from 30 selected plants along the harvesting phase, i.e., approximately 260 peppers from OA and 370 from CA were collected. (lines 347-49).
remark
Line 209 ‘In the projection are presented the 5 samples’ should be changed to: Five samples are presented in the projection’
- Your remark was performed.
remark
Line 297:’ Excessive intake of P per se can be harmful to bones through increased parathyroid secretion, but the adverse effects on bone mass increase, when dietary Ca intake is clearly low.’ – a reference is desirable
- The reference was placed in the end of the sentence [35] – line 275
remark
Line 312 ‘ratios does not exceeded’ should be changed to “did not exceed’
- Your remark was performed. (line 288)
Author´s Concluding Remark
The discussion was shortened despite new arguments were added. For example in the last draft the Discussion contained 127 lines and the current one, only 110 lines including 12 new lines (248-55; 264-67).
The phenological development stages of sweet pepper from OA and CA during 2018 was also included, see lines 338-42 and Table 5 (lines 358-60).
Best Regards
Fernando H. Reboredo
Round 2
Reviewer 2 Report
The authors have made important changes that improved the quality of the manuscript. Although, it still have some important gaps that make its publication non-viable in the present form. As I stated in the previous review, I do not considered that the information reported have enough relevance for being published in a high quality journal as Plants-MPDI, but I leave the final decision to the assistant editor.
Here you can find some comments that it should be considered:
L- 44. Keywords: Please, do not repeat words that appeared in the title (i.e. elemental composition …)
L 416. Said here the duration of the experiment. From plantation to harvest (months??) as you specify in Table 5.
L 417. You report that both farms (OA and CA) grown an open-air, then, how was the distance between them? In this sense, you should specify in this section (4.1). the meteorological conditions registered in the area, maximum temperature, evapotranspiration, rainfall… that might influence the ripening stages.
L431. Specify the herbicide used (concentration), and the moment of application.
I still don´t understand the experimental layout used and the samples considered for each determination. This is wordy explained. For example, in L437-443 some information is repeated in section 4.2 and 4.4.
You said ‘ Four independent batches with approximately 2 kg each (2 for each production mode), were 440 formed, and then each batch was sub-divided in five parts’ and then you also said: 10 sweet peppers of each production mode were chosen, (5 replicates for each ripening stage) in the total of 20 sample, and in other lines ‘ after the determination of dry matter and 450 moisture, five replicates of each sample (powder reduced) were used for XRF analysis’
I recommend to specify more clear the samples collected and used for each determination. Maybeyou can used a separated section.
Furthermore, the equipment, methodology and samples used for physico-chemical quality that appear in Table 2. is omitted. The reference of Barroca et al., 2020 is not enough. Please, specify. In this sense.....the authors must specify how they obtained all the results presented in the manuscript.
Section 4.5. Ok, But you must include the software used.
L 250- 256 PCA conclusions here? You must merged this information with section 4.5. Or maybe rewritten this information in one only paragraph.
L 273. Change ‘by us’. Its too colloquial. You can used: results/work/experiment or study.
L 286. Check the references format to the authors instructions.
In the section 5 of conclusions, you must avoid to repeat results with data.
Author Response
COVER LETTER
Manuscript ID: plants-822216
REVIEWER 2
Here you can find some comments that it should be considered:
L- 44. Keywords: Please, do not repeat words that appeared in the title (i.e. elemental composition …)
L 416. Said here the duration of the experiment. From plantation to harvest (months??) as you specify in Table 5.
L 417. You report that both farms (OA and CA) grown an open-air, then, how was the distance between them? In this sense, you should specify in this section (4.1). the meteorological conditions registered in the area, maximum temperature, evapotranspiration, rainfall… that might influence the ripening stages.
L431. Specify the herbicide used (concentration), and the moment of application.
I still don´t understand the experimental layout used and the samples considered for each determination. This is wordy explained. For example, in L437-443 some information is repeated in section 4.2 and 4.4.
You said ‘ Four independent batches with approximately 2 kg each (2 for each production mode), were 440 formed, and then each batch was sub-divided in five parts’ and then you also said: 10 sweet peppers of each production mode were chosen, (5 replicates for each ripening stage) in the total of 20 sample, and in other lines ‘ after the determination of dry matter and 450 moisture, five replicates of each sample (powder reduced) were used for XRF analysis’
I recommend to specify more clear the samples collected and used for each determination. Maybeyou can used a separated section.
Furthermore, the equipment, methodology and samples used for physico-chemical quality that appear in Table 2. is omitted. The reference of Barroca et al., 2020 is not enough. Please, specify. In this sense.....the authors must specify how they obtained all the results presented in the manuscript.
Section 4.5. Ok, But you must include the software used.
L 250- 256 PCA conclusions here? You must merged this information with section 4.5. Or maybe rewritten this information in one only paragraph.
L 273. Change ‘by us’. Its too colloquial. You can used: results/work/experiment or study.
L 286. Check the references format to the authors instructions.
In the section 5 of conclusions, you must avoid to repeat results with data.
Author´s Remark
All your comments are taken into account and new lines (blue) were added to the manuscript in order to differentiate from the others.
A new table (Table 5) was also introduced as requested in your remark L417.
Some particular remarks I would like to emphasize and I agree with you that the sampling description is not understandable for the audience.
In this context we follow your suggestion and we create a separate section (4.2) called – Sample collection. In this section we explained in a more concise way the total number of samples and how they are divided.
Your request about methods and equipment regarding firmness, TA and TTS are referred - lines 378-391. I avoid to include the formula for TA determination, but if necessary it will be added.
Another point I would like to stress is related with the PCA analysis. Talking with my colleague responsible for such task he give me the advice to insert the PCA in the point 4.5 Statistical analysis, with the appropriate reference to the Software used as requested by you.
But if you think that it is better to maintain a separate item of PCA analysis and another item so-called Statistical analysis, we will do it.
PCA data from conclusion was removed as well as data from elemental composition.
We want to express our gratitude to you, and all the Reviewers, for such advices that clearly improved the manuscript.
Best Regards
Fernando H. Reboredo

Round 3
Reviewer 2 Report
Thanks for incorporating all the suggestions made.
I don't feel qualified to judge about the English language and style, but maybe the manuscript need to be revised by a native English speaker since some sentences need to be rephrased in a straigthforward way
The table 5 is not necessary. Just add the average values of the experimental period to the text (L327) of each meterological variable.
Author Response
COVER LETTER
Manuscript ID: plants-848192
REVIEWER 2
Remark
The table 5 is not necessary. Just add the average values of the experimental period to the text (L327) of each meterological variable.
This comment was changed as requested - see lines 330-335 (blue lines)
Best regards
Fernando H. Reboredo

This manuscript is a resubmission of an earlier submission. The following is a list of the peer review reports and author responses from that submission.
Round 1
Reviewer 1 Report
Introduction provides a lot of information on organic agriculture. This kind of information could be substantially decreased, in order to reduce the length of the Introduction.
In some cases, there is a monotonous description of all the results, without deepening into a thorough Discussion. I advise the authors to decrease these descriptive parts and deepen into a more constructive comparison of their data to similar of other researchers.
I recommend the Editor to reconsider the paper for publication after a satisfactory moderate review.
Reviewer 2 Report
This manuscript investigated the nutritional composition of pepper from different agricultural production systems (conventional and organic).
In general, the manuscript is well written, the approach is well explained and the objectives are clear. However, in my opinion, the study is insufficient. In order to evaluate a comparison between two different agricultural systems, a more in-depth study could have been done, since the manuscript only provides results of mineral content and some nutritional characterization. Information regarding production (kg/ha, kg/pl, fruit average weight…) and quality parameters such as ºBrix, acidity, firmness and fruit colour would have provided useful information to complete the study. On the other hand, the conditions exposed in the two fields are not clear and some information is missing. Authors only describe that soil characteristics were similar in both fields and they were irrigated by drip irrigation. But, was the type of water and quantity the same in both situations? How many hectares did each field have? Was the same density of plants applied in both fields? It could have been interesting to see a table with the quantity of nutrients provided by each type of fertilization and a table with the different cultural practices applied in each case, including the application of agrochemicals or organic products.
Based on the above, I recommend this manuscript to be rejected.
Reviewer 3 Report
All about this work, the questions, suggestions and comments were listed on the reviewed pdf document that attached in the system.
